# How Good is my Video LMM? Complex Video Reasoning and Robustness Evaluation Suite for Video-LMMs

## Abstract

Recent advancements in Large Language Models (LLMs) have led to the development of Video Large Multi-modal Models (Video-LMMs) that can handle a wide range of video understanding tasks. These models have the potential to be deployed in real-world applications such as robotics, AI assistants, medical surgery, and autonomous vehicles. The widespread adoption of Video-LMMs in our daily lives underscores the importance of ensuring and evaluating their robust performance in mirroring human-like reasoning and interaction capabilities in complex, real-world contexts. However, existing benchmarks for Video-LMMs primarily focus on general video comprehension abilities and neglect assessing their reasoning capabilities over complex videos in the real-world context, and the robustness of these models through the lens of user prompts as text queries. In this paper, we present the Complex Video Reasoning and Robustness Evaluation Suite (CVRR-ES), a novel benchmark that comprehensively assesses the performance of Video-LMMs across 11 diverse real-world video dimensions. We evaluate 11 recent models, including both open-source and closed-source variants, and find that most of the Video-LMMs, especially open-source ones, struggle with robustness and reasoning when dealing with complex videos. Based on our analysis, we develop a training-free Dual-Step Contextual Prompting (DSCP) technique to effectively enhance the performance of existing Video-LMMs on CVRR-ES benchmark. Our findings provide valuable insights for building the next generation of human-centric AI systems with advanced robustness and reasoning capabilities. Our dataset and code will be made publicly available.

## 1 Introduction

Large Language Models (LLMs) (Touvron et al., 2023; Zheng et al., 2023; Jiang et al., 2024) have recently demonstrated emerging reasoning and planning capabilities. These models can simultaneously solve a wide array of natural language processing (NLP) tasks, including summarization, QA, and machine translation (Wei et al., 2022a; Brown et al., 2020). Consequently, their integration with the vision modality, specifically for video understanding tasks, has given rise to Video Large Multimodal Models (Video-LMMs) (Li et al., 2023b). These models act as visual chatbots that accept both text and video as input and handle a diverse set of tasks, including video comprehension (Maaz et al., 2023), detailed video understanding (Lin et al., 2023), and action grounding (Zhang et al., 2023). As these models directly capture video data, they hold substantial potential for deployment in real-world applications such as robotics, surveillance, medical surgery, and autonomous vehicles.

However, as these models assume an expanding role in our everyday lives, assessing their performance in comprehending complex videos and demonstrating reliable reasoning and robustness capabilities across diverse real-world contexts becomes essential. Video-LMMs with such capabilities will be more effective when integrated into our daily lives for solving perception tasks and will be a promising step towards building trustworthy human-centric AI-assistive systems (OpenAI, 2024).

Several attempts in literature have been made to benchmark Video-LMMs. SEED-Bench (Li et al., 2023a) curated a MCQ-based dataset including 3 evaluation dimensions for videos. Similarly, MV-Bench (Li et al., 2023c) constructed the Video-LMM benchmark and assembled 20 video tasks for

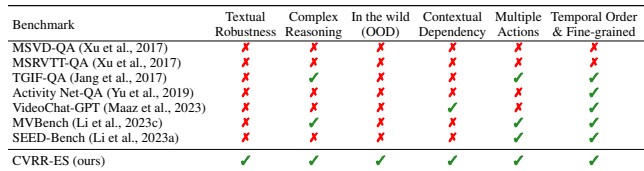

| Benchmark | Textual Robustness | Complex Reasoning | In the wild (OOD) | Contextual Dependency | Multiple Actions | Temporal Order & Fine-grained |
|---|---|---|---|---|---|---|
| MSVD-QA (Xu et al., 2017) | ✗ | ✗ | ✗ | ✗ | ✗ | ✗ |
| MSRVTT-QA (Xu et al., 2017) | ✗ | ✓ | ✗ | ✗ | ✓ | ✓ |
| TGIF-QA (Jang et al., 2017) | ✗ | ✓ | ✗ | ✗ | ✓ | ✓ |
| Activity Net-QA (Yu et al., 2019) | ✗ | ✗ | ✗ | ✗ | ✗ | ✓ |
| VideoChat-GPT (Maaz et al., 2023) | ✗ | ✗ | ✗ | ✓ | ✗ | ✓ |
| MVBench (Li et al., 2023c) | ✗ | ✓ | ✗ | ✗ | ✓ | ✓ |
| SEED-Bench (Li et al., 2023a) | ✗ | ✗ | ✗ | ✗ | ✓ | ✓ |
| CVRR-ES (ours) | ✓ | ✓ | ✓ | ✓ | ✓ | ✓ |

Table 1: Comparison of CVRR-ES with existing benchmarks for video question answering. The CVRR-ES benchmark represents an initial effort to assess Video-LMMs in the context of their applicability and suitability in real-world contexts.

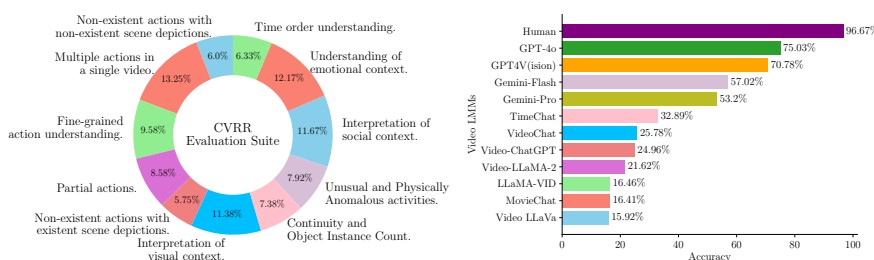

Figure 1: **Left:** CVRR-ES comprises of 11 diverse complex video evaluation dimensions encompassing a variety of complex, real-world contexts. **Right:** Overall performance of Video-LMMs on the CVRR-ES benchmark. Results for each Video-LMM are averaged across 11 video dimensions.

evaluating the spatial and temporal understanding of these models. While these methods aim at benchmarking Video-LMMs, they predominantly evaluate video and/or temporal comprehension abilities and overlook the complex reasoning aspects of Video-LMMs for real-world context, and their robustness towards user input text queries; both of which are crucial to ensure their responsible engagement with humans in various real-world situations in the wild. While some studies have explored similar areas such as hallucinations in image-based LLMs (Liu et al., 2023a; Qian et al., 2024), no such comprehensive study exists for the case of Video-LMMs.

Motivated by the wide-scale applications of Video-LMMs and the lack of world-centric complex video benchmarking efforts, we present a new benchmark, Complex Video Reasoning and Robustness Evaluation Suite (CVRR-ES), to comprehensively assess the performance of Video-LMMs. As shown in Tab. 1, CVRR-ES evaluates Video-LMMs on key aspects of robustness and reasoning in videos, encompassing video domains that more accurately test models in real-world scenarios such as videos having contextual dependency and in-the-wild aspects. CVRR-ES is an open-ended video QA benchmark comprising 11 real-world video category dimensions (Fig. 1, left) that encompass diverse evaluation aspects. These dimensions span from context-dependent (e.g., social, emotional, etc.) categories to ones that often take place in the wild such as videos containing physically anomalous activities. We comprehensively evaluate a representative set of 11 recent Video-LMMs (Fig. 1, right) including both open-source and closed-source models on the CVRR-ES benchmark using a LLM-assisted automatic evaluation framework (Maaz et al., 2023; Cai et al., 2023).

The performance of Video-LMMs on the CVRR-ES benchmark reveals that these models struggle to correctly comprehend complex videos indicating their weak reasoning and lack of robustness to the textual user queries (Fig. 2). For instance, state-of-the-art Video-LLaVA (Lin et al., 2023) achieves only 15.92% performance averaged across 11 video dimensions of CVRR-ES. In contrast, closed-source models including GPT4V(vision) (OpenAI, 2023) and Gemini-Vision-Pro (Google, 2023) exhibit relatively stronger performance but still lag behind the performance of humans. Using CVRR-ES benchmark, we extensively perform quantitative and qualitative analysis and formulate important insights about these Video-LMMs based on their failure cases and individual performances across the diverse video dimensions.

Our thorough analysis show that standard prompting struggles in steering Video-LMMs' focus for complex video understanding. Additionally, their limitations in reasoning and robust video understanding of real-world scenarios are dominantly driven by the quality of textual inputs (i.e., user questions). Based on these insights, we develop a training-free Dual-Step Contextual Prompting (DSCP) technique, which effectively steers the model's behavior during inference to elicit video-specific reasoning and improved robustness in Video-LMMs. With DSCP, Video-LMMs substantially improve on our benchmark, suggesting the potential of prompting methods for Video-LMMs.

Our main contributions are as follows: (1) We present Complex Video Robustness and Reasoning Evaluation suite (CVRR-ES), a Video Question Answering benchmark designed to assess the reasoning and robustness capabilities of Video-LMMs on 11 diverse world-centric complex video dimensions (§3). (2) We extensively evaluate both open-source and closed-source Video-LMMs on the CVRR-ES benchmark and find that most models exhibit weak performance, highlighting their

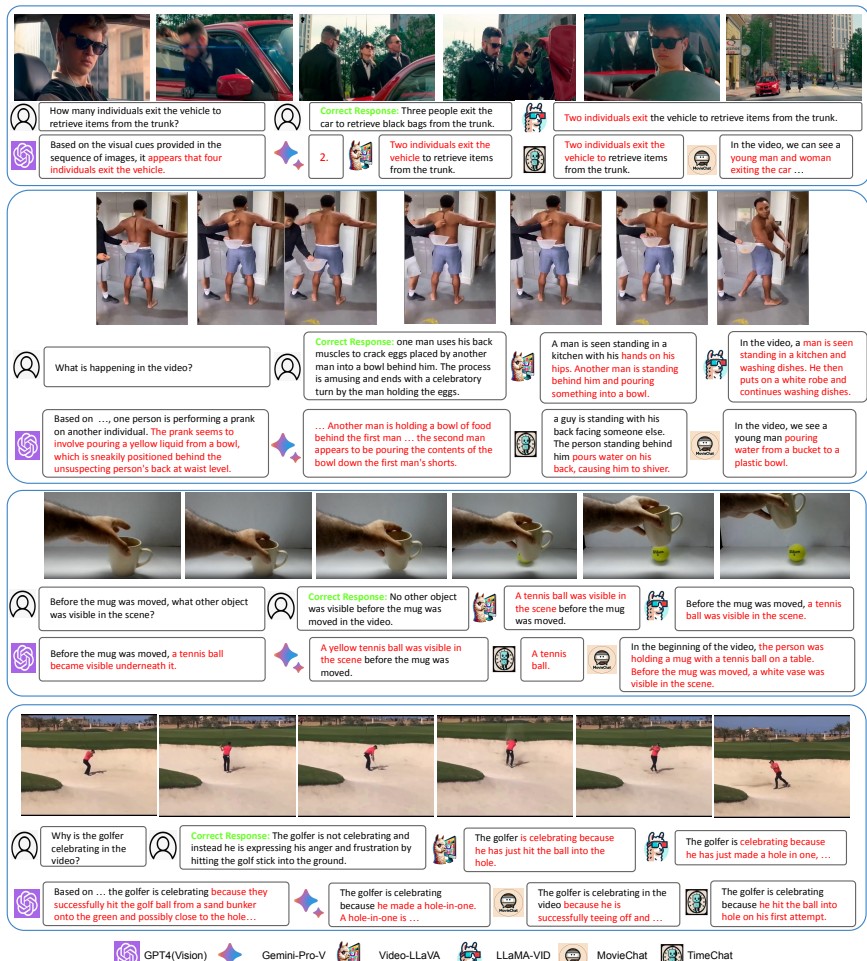

Figure 2: We observe that most Video-LMMs struggle to reason over complex videos (rows 1-3) and exhibit weak robustness and rectification abilities when answering user questions that can sometimes be confusing (row 4). The QA pairs in Comprehensive Video Reasoning and Robustness Evaluation Suite (CVRR-ES) benchmark assess the performance of Video-LMMs beyond general video comprehension. (best viewed zoomed in)

limited reasoning in complex videos and lack of robustness towards user text queries (§5.1). (3) We conduct comprehensive analysis and formulate important conclusions about Video-LMMs based on their failure cases and performance on the CVRR-ES. Our findings provide key insights for building the next generation of human-centric AI systems with improved robustness and reasoning abilities (§5.2). (4) To improve Video-LMMs' reasoning and robustness abilities, we design a model-agnostic and training-free prompting method that effectively enhances their performance (§4).

## 2 RELATED WORKS

**Video Large Multi-modal models (Video-LMMs).** Video-LMMs (Lin et al., 2023; Li et al., 2023d; Zhang et al., 2023) are visual chatbots capable of performing a wide range of video tasks, including video comprehension and captioning, video question-answering, and action grounding. These models accept both video and textual inputs and generate textual responses. From an architectural perspective, Video-LMMs combine pre-trained vision backbones (Radford et al., 2021; Fang et al., 2023; Wang et al., 2022b) with large language models (Touvron et al., 2023; Zheng et al., 2023) using connector modules such as MLP adapters, Q-former (Dai et al., 2023), and gated attention (Alayrac et al., 2022). VideoChat (Li et al., 2023b) and VideoChat-GPT (Li et al., 2023d) presented initial open-source efforts in this direction and were trained with two stages of alignment and video-instruction following objectives. Recently, more advanced Video-LMMs have emerged in the field, with some models focusing on improving model architectures (Li et al., 2023d), expanding to new tasks (Munasinghe et al., 2023), and enabling support for long videos (Song et al., 2023; Ren et al., 2023). In this work, we aim to develop a comprehensive benchmarking framework to assess the reasoning and robustness capabilities of these Video-LMMs and develop a training-free prompting technique to improve their performance on these fronts.

**Benchmarking Video-LMMs.** With the growing number of Video-LMMs emerging in the research community, several works have presented evaluation frameworks to assess and quantify these models for benchmarking and analysis purposes. SEED-Bench (Li et al., 2023a) evaluates the visual capabilities in both image and Video-LMMs across 12 unique dimensions. MV-Bench (Li et al., 2023c) curates 20 video tasks to evaluate the spatial and temporal understanding of Video-LMMs. Video-Bench evaluates Video-LMMs across 10 tasks in three areas: Video-exclusive, Prior Knowledge, and Decision-making. TempCompass (Liu et al., 2024) introduces a temporally challenging benchmark with conflicting videos sharing static content but differing in temporal aspects. Auto-Eval-Video (Chen et al., 2023) creates open-ended video QAs across nine dimensions. Video-Bench (Ning et al., 2023) evaluates Video-LMMs across 10 tasks in three areas: Video-exclusive, Prior Knowledge, and Decision-making. VALUE (Li et al., 2021) focuses on Video QA, retrieval, and Video-captioning tasks. While these evaluation frameworks provide effective insights, their assessments do not extend beyond general video-comprehension metrics to more advanced aspects of reasoning and robustness, particularly for real-world context cases. In this work, we conduct initial research efforts on providing a complex video reasoning and robustness benchmark and offer a thorough assessment of Video-LMMs in practical and in-the-wild scenarios.

**Training-free Prompting Techniques.** Steering model behavior at inference time using prompting has become a common paradigm in the NLP domain. Prompting (Wei et al., 2022b; Wang et al., 2022a) refers to the set of instructions given as a prefix to the language model to better align model responses with human intent without the need for task-specific fine-tuning. Prompting techniques can be as simple as a single sentence (e.g., "Let's think step by step") such as zero-shot chain of thought (Wei et al., 2022b) prompting, to more detailed techniques such as combining chain-of-thought prompting with few-shot learning (Brown et al., 2020) and self-consistency chain of thought prompting (Wang et al., 2022a). Surprisingly, training-free prompting techniques for Video Large Multi-modal Models (Video-LMMs) have been minimally explored. In this work, we develop a dual-step prompting technique based on principled prompt instructions specifically designed to steer the model's behavior for improved reasoning and robustness over complex videos.

## 3 COMPLEX VIDEO REASONING AND ROBUSTNESS EVALUATION SUITE

As Video-LMMs are touching new real-world applications, it is essential to ensure that they robustly handle the user inputs, comprehend the visual world, and exhibit human-like reasoning capabilities. In this work, our goal is to establish a comprehensive benchmark, Complex Video Reasoning and Robustness Evaluation Suite (CVRR-ES) to assess the *robustness* and *reasoning* capabilities of Video-LMMs over complex and real-world contextual videos. We first provide an overview of CVRR-ES and then detail the video evaluation dimensions in Sec. 3.1. Subsequently, we discuss the benchmark creation process in Sec. 3.2.

**Overview.** CVRR-ES encompasses evaluation dimensions that cover diverse video categories related to real-world scenarios, ranging from context-dependent (e.g., social, emotional) categories to video types that often take place in the wild (e.g., anomalous activities). Specifically, we have compiled 11 video evaluation dimensions and curated 2,400 high-quality open-ended question-answer (QA) pairs, spanning 214 high-quality videos. The average video duration is 22.3 seconds, with maximum and minimum durations of 183 and 2 seconds, respectively. Fig. 2 shows qualitative examples of the collected videos for the CVRR-ES benchmark. Refer to Appendix D for additional statistical details and qualitative results.

### 3.1 CVRR-ES VIDEO CATEGORY DEFINITIONS.

For curating the CVRR-ES benchmark, we carefully select 11 diverse benchmark evaluation categories. As shown in Fig. 1 (left), these categories encompass a wide range of real-world complex and contextual video types. Below, we define each video evaluation dimension in detail.

**1) Multiple actions in a single video.** This category involves videos with 2-4 different human activities. We curate questions in this category to assess the model's ability to understand and reason about multiple actions and their interrelations in a single video.

**2) Fine-grained action understanding.** We collect videos that encompass fine-grained activities performed by humans, such as pushing, opening, closing, spreading, sitting, etc. This category tests the model's ability to interpret subtle and fine-grained actions through carefully crafted questions.

**3) Partial actions.** We observe that Video-LMMs generate content that is relevant to a video's context and likely to occur next. We collect videos with actions likely to be followed by other actions but not triggered in the video e.g., cracking an egg in a kitchen suggests the next action of cooking the egg. This dimension assesses Video-LMMs on their ability to correctly identify partial actions.

**4) Time order understanding.** Accurately recognizing the temporal sequence of activities in videos is crucial for distinguishing between atomic actions, such as pushing and pulling. We collect videos of fine-grained actions occurring in a particular temporal direction and curate challenging questions.

**5) Non-existent actions with existent scene depictions.** This category examines the model's robustness and reasoning behavior in scenarios where we introduce non-existent activities into the video without altering the physical and spatial scenes or environmental details in it.

**6) Non-existent actions with non-existent scene depictions.** In this category, we increase the difficulty of the QA task by including questions containing both non-existent activities and scenes. We alter the details of objects, attributes, and background for non-existent scene comprehension. This tests the model's ability to correct misleading questions and avoid generating imaginary content.

**7) Continuity and object instance count.** We curate videos (real-world and simulations) designed to test the models' ability to accurately recognize the number of instances of objects, people, etc., and distinguish between existing objects and new ones introduced later in the same video scene.

**8) Unusual and physically anomalous activities.** We collect videos depicting unusual actions that seemingly defy the laws of physics, such as a person floating in the air or driving a motorbike on a running river. Assessing Video-LMMs in such fronts is crucial, as it allows us to determine whether they can generalize to understand actions in out-of-distribution videos in practical situations.

**9) Interpretation of social context.** We test Video-LMMs' ability to understand actions influenced by social contexts, such as helping an elderly person cross the road. Video-LMMs are assessed to determine their ability to accurately infer the rationale behind actions using the social context.

**10) Understanding of emotional context.** Similar to social context, humans can accurately understand and interpret each other's actions by considering the emotional context. We test Video-LMMs' ability to understand actions based on emotional context, e.g., a person crying due to joy.

**11) Interpretation of visual context.** This category tests the model's ability to understand actions by leveraging the overall visual contextual cues in the video. For example, to identify the number of people present based on the presence of shadows, one must utilize the visual context of shadows.

### 3.2 BUILDING CVRR-ES BENCHMARK

**Stage 1: Data collection and Human Annotations.** We first collect high-quality videos and annotate each video via human assistance. To ensure that each evaluation dimension captures relevant attributes and information, we meticulously select videos that are representative of specific characteristics associated with that dimension. Overall, 214 unique videos are selected covering 11 dimensions with around 20 videos per evaluation dimension. Around 60% of these videos are collected from public academic datasets while respecting the distribution rights of the original datasets. To introduce diversity in the benchmark distribution, we select videos from multiple datasets including Something-Something-v2 (Goyal et al., 2017), CATER (Girdhar & Ramanan, 2020), Charades (Sigurdsson et al., 2016), ActivityNet (Caba Heilbron et al., 2015), HMDB51 (Kuehne et al., 2011), YFCC100M (Thomee et al., 2016). The remaining 40% of videos are collected from internet. Following the video collection process, two human annotators (authors of this work) generate captions for each video. For videos where initial captions or metadata are available from academic datasets, the captions are generated by the annotators based on them. For videos collected from the internet, captions are entirely generated by annotators. To ensure consistency and high quality, we provide annotation instructions to annotators, who generate captions accordingly. Personalized annotation guidelines are used for each video category. Refer to additional details in Appendix D.

**Stage 2: Question-Answer Generation.** The first challenge is to select an evaluation setting to assess Video-LMMs. Humans typically engage in free-form conversation to interact with each other in day-to-day life. Inspired by this, we aim to simulate a similar style of interaction with Video-LMMs by curating open-ended QA pairs to evaluate these models for robustness and reasoning. We feed detailed ground-truth video captions to GPT-3.5 LLM, which is utilized to generate open-ended questions. The QA pairs covers both the reasoning and robustness aspects as detailed below.

**Reasoning QA pairs:** With Video-LMMs beginning to interact more directly with humans in our lives, it's crucial to validate the reasoning abilities of Video-LMMs for more reliable Human-AI interaction. When evaluating the reasoning capabilities of Video-LMMs, we define complex video reasoning as a two-step process; i) The model's ability to understand a video not only by analyzing spatial content but also by grasping the underlying rationale behind the occurring activities and their relationships with surrounding context. ii) Relating the retrieved contextual interpretation to the user's query to provide a grounded response. This involves creating questions that go beyond simple video comprehension and scene description and require the model to engage in complex logical inference, contextual understanding, and reasoning about counterfactual and hypothetical scenarios.

This includes providing reliable answers to questions such as 'How,' 'Why,' 'Is,' and 'When,' which require contextual understanding, and Video-LMM's ability to decode counterfactual scenarios.

**Robustness QA pairs:** In addition to evaluating the reasoning capabilities of LLMs, it is important to assess Video-LMMs to ensure their robust and responsible performance in real-world scenarios. In the context of Video-LMMs, robustness can be evaluated from both visual (video input) and textual interfaces. Our focus in this work lies on textual interface robustness by particularly gauging the resilience and self-correcting abilities of Video-LMM when there is variability in user text queries; particularly with wrong, confusing, and misleading cues. This scenario mirrors realistic situations where users, based on their expertise levels, may pose irrelevant, misleading, or confusing questions. It is crucial for models to demonstrate reliability and robustness in handling such queries and avoid generating unreal or hallucinated content for input videos.

We curate specific prompts for each evaluation dimension to instruct LLM in generating QA pairs. Example prompts used as an instruction to LLMs for curating QA pairs for robustness and reasoning aspects are provided in Fig. 15 in the Appendix F.

**Stage 3: QA Pairs Filtration via Human Verification.** After generating the QA pairs, we employ a manual filtration step, with human assistance to verify each generated QA pair. Approximately 30% of the QA pairs generated by GPT-3.5 are found to be noisy, containing questions that are unrelated to the video evaluation dimensions or unanswerable based on the provided ground-truth captions. Additionally, many questions are repetitive and contain answers within the question itself. Therefore, an exhaustive filtering process is conducted which involves QA rectification and removing those samples which are not relevant to the video or evaluation type. This process results in a final set of 2400 high-quality QA pairs for the CVRR-ES benchmark. Examples of the final QA pairs are shown in Tab. 4 in the Appendix. This stage further ensured the diversity of QA pairs.

**Stage 4: Evaluation Procedure.** Previous methods in the literature (Maaz et al., 2023; Cai et al., 2023; Liu et al., 2023a; Qian et al., 2024) have explored using LLM models as judges for quantifying results in open-ended QA benchmarks. We adopt a similar approach and instruct LLMs to act as teachers to assess the correctness of predicted responses from Video-LMMs compared to ground-truths. We generate open-ended predictions from Video-LMMs by providing video-question pairs as inputs and then present the model predictions and their ground-truth responses to the LLM Judge using the evaluation prompt. The Judge determines whether the prediction is correct or incorrect with a binary judgment, assigns a score from 1 to 5 representing the quality of the prediction, and provides a reasoning to explain its decision. Our ablative analysis in the Appendix. F demonstrates that reasoning-constrained LLM-based evaluation aligns the most with human-based judgment. Our evaluation prompt for LLM Judge is shown in Fig. 14 in Appendix F.

**Quality of QA pairs.** We show examples of QA pairs from CVRR-ES benchmark in Table 4 in Appendix D. Our QA pairs are of high quality and aim to test the understanding of Video-LMMs against reasoning and robustness criteria on multiple evaluation dimensions. To quantitatively assess the quality of the benchmark, we establish a quality assessment procedure (Gandhi et al., 2024). We randomly sample 1120 QA pairs, which encompass all videos of the CVRR-ES benchmark, and request human experts to evaluate the quality of each QA pair by answering the following questions: **(1)** "*Does the QA pair correctly represent the evaluation dimension category under which it falls?*" (possible answers: "Yes", "No") **(2)** *Can the question be correctly answered given only the video content?* (possible answers: "Agree", "Disagree") and **(3)** *Is the corresponding paired ground-truth answer correct? (which will be used during evaluation as ground truth)* (possible answers: "Yes", "No"). On average, the answer of experts for the first question was "**Yes**" for 98.84% of the times. For the second and third questions, the averaged answer was "**Agree**" and "**Yes**" for 100% and 99.91% of the times, respectively.

**Human Evaluation.** To verify that the QA pairs in the CVRR-ES benchmark are reasonably answerable and to establish a benchmark for human performance, we conduct a human evaluation. Two human experts (authors) are instructed to watch the video corresponding to each question and provide a free-form answer. The predictions of the human experts for all QA pairs are assessed using an LLM-assisted evaluation. Individual final scores are averaged to mitigate potential bias from a single human evaluator. Human evaluation results are presented in experiments section (Sec. 5.1).

# 4 DUAL-STEP CONTEXTUAL PROMPTING FOR VIDEO-LMMS.

Given their wide-scale potential in practical applications, new Video-LMMs are frequently introduced by the research community. Despite the availability of numerous Video-LMMs, the majority of them are trained using only positive examples and video-conversational templates that are primarily limited to tasks such as video-captioning and video question answering (Li et al., 2023b; Maaz

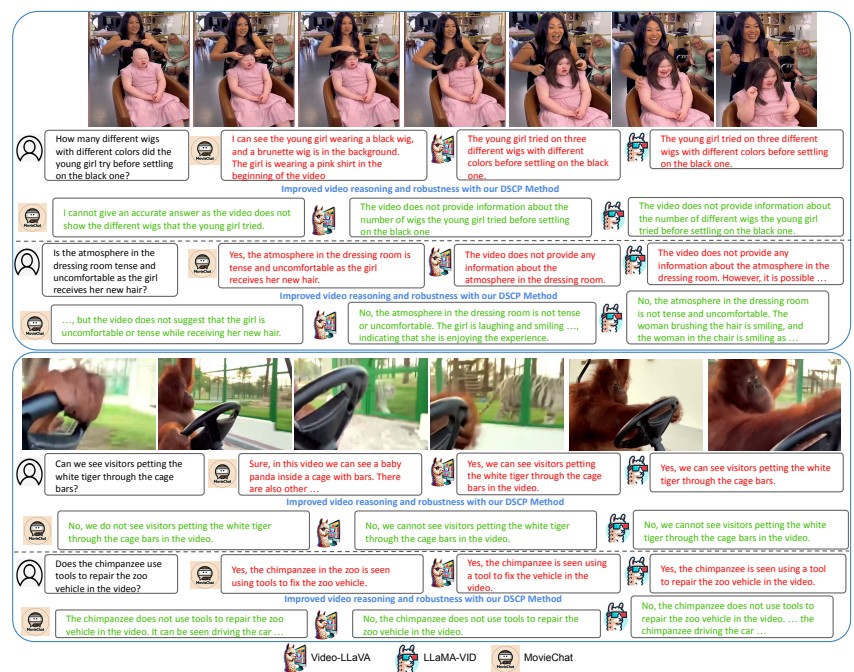

Figure 3: **Qualitative results of DSCP prompting method.** Using our DSCP approach, Video-LMMs demonstrate enhanced robustness and reasoning capabilities over complex videos.

et al., 2023; Ren et al., 2023; Song et al., 2023). This leads to highly over-affirmative behavior and a lack of self-rectification abilities in these models (Sec. 5.2).

Additionally, the templates have minimal focus on enhancing reasoning and robustness capabilities through reasoning instruction-tuning pairs, resulting in their weak performance against robustness and reasoning based evaluations in CVRR-ES. Consequently, enabling direct interaction of Video-LMMs with users in real-world scenarios can result in undesired responses when the user question is confusing and deceiving. Moreover, curating reasoning-based instruction fine-tuning datasets requires meticulous data curation steps, and retraining these models are computationally expensive (Li et al., 2023d; Ren et al., 2023). Alternatively, training-free prompting techniques in NLP literature have shown effectiveness in eliciting reasoning abilities in LLMs such as chain of thought and self-consistency prompting (Wei et al., 2022b; Wang et al., 2022a). Inspired by these, we present a Dual Step Contextual Prompting (DSCP) technique, which steers Video-LMM focus for enhanced reasoning while simultaneously encouraging the models to provide robust and grounded answers. DSCP is a two-step prompting method that **1)** ensures that the model comprehends the video while reasoning over crucial aspects of complex video understanding such as contextual information and decoding the complex relationships between objects and motions, etc., and **2)** encourages robustness by generating the response against the question while conditioning both on video and the unbiased context retrieved in the first step. Below we discuss each step of DSCP in detail. **Step 1: Video reasoning.** We prompt Video-LMMs to interpret video from a reasoning perspective using ten principled instructions (Fig. 4, in **blue**) to direct the models to understand general video content, reason over the rationale behind actions and their relationships with the context, and consider factors like contextual priors, the temporal order of actions, instance count, and attributes. The prompting technique also includes instructions to ensure conciseness and factuality to mitigate hallucinations. Given a Video-LMM $\mathcal{F}$ and input video $\mathcal{V}$, we retrieve contextual reasoning information $I_{\text{context}}$ by providing principled reasoning prompt $P_{\text{reason}}$ along with the video

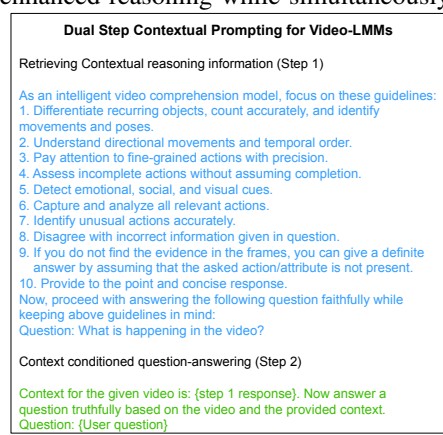

Figure 4: Principled prompt instructions in DSCP for Video-LMMs.

Table 2: Evaluation results of Video LLMs across various video-evaluation categories on the CVRR-ES benchmark. We present results for both open-source and closed-source models and human evaluation.

| Benchmark Category | Video-LLaMA-2 | VideoChat | Video-ChatGPT | Video-LLaVA | MovieChat | LLaMA-VID | TimeChat | Gemini-V Pro | Gemini-V Flash | GPT4V | GPT4o | Human |
|---|---|---|---|---|---|---|---|---|---|---|---|---|
| Multiple Actions in single video. | 16.98 | 23.90 | 27.67 | 15.72 | 12.58 | 17.92 | 28.30 | 43.08 | 44.65 | 57.55 | 62.89 | **93.40** |
| Fine-grained action understanding. | 29.57 | 33.48 | 26.96 | 25.22 | 23.48 | 26.09 | 39.13 | 51.61 | 64.78 | 77.39 | 80.43 | **95.65** |
| Partial actions. | 24.76 | 33.01 | 22.82 | 13.59 | 21.36 | 14.56 | 49.51 | 67.48 | 62.14 | 73.79 | 77.67 | **98.54** |
| Time order understanding. | 16.45 | 31.58 | 27.63 | 21.05 | 16.45 | 19.74 | 34.21 | 45.39 | 55.26 | 57.89 | 71.05 | **97.37** |
| Non-existent actions with existent scene. | 10.14 | 15.22 | 23.19 | 5.07 | 5.07 | 2.90 | 23.19 | 57.25 | 60.14 | 71.01 | 83.33 | **97.10** |
| Non-existent actions with non-existent scene. | 13.19 | 14.58 | 17.36 | 3.47 | 11.81 | 6.94 | 13.89 | 49.64 | 56.30 | 75.00 | 70.14 | **100.00** |
| Continuity and Object instance Count. | 28.25 | 24.29 | 28.41 | 21.47 | 19.77 | 24.86 | 34.46 | 36.16 | 43.50 | 62.71 | 62.71 | **96.49** |
| Unusual and Physically Anomalous activities. | 18.95 | 18.42 | 18.95 | 15.79 | 17.89 | 16.32 | 27.37 | 60.00 | 60.53 | 74.74 | 78.42 | **96.84** |
| Interpretation of social context. | 25.00 | 31.07 | 32.50 | 18.93 | 17.14 | 13.93 | 39.29 | 64.29 | 69.64 | 79.64 | 83.57 | **97.51** |
| Understanding of emotional context. | 21.92 | 23.63 | 21.23 | 15.07 | 13.70 | 14.73 | 27.40 | 47.26 | 52.74 | 66.44 | 70.89 | **95.55** |
| Interpretation of visual context. | 32.60 | 34.43 | 27.84 | 19.78 | 21.25 | 23.08 | 45.05 | 63.00 | 57.51 | 82.42 | 84.25 | **94.87** |
| **Average** | 21.62 | 25.78 | 24.96 | 15.92 | 16.41 | 16.46 | 32.89 | 53.20 | 57.02 | 70.78 | 75.03 | **96.67** |

| Prompting Method | VideoChat | Video-LLaVA | MovieChat | LLaMA-VID | TimeChat |
|---|---|---|---|---|---|
| Standard prompting | 25.78 | 15.92 | 16.41 | 16.46 | 32.89 |
| Chain of Thought (CoT) prompting | 22.44 | 25.87 | 15.89 | 29.68 | **39.57** |
| DSCP (Stage 1) | 38.07 | 32.12 | 28.05 | 25.13 | 33.04 |
| DSCP (Both stages) | **47.92** | **37.93** | **35.87** | **46.85** | 39.45 |

Table 3: **Prompting methods.** DSCP stage 1 uses only principled instructions of step 1 and DSCP (Both stages) uses a complete dual-step technique.

to the LMM, $I_{\texttt{context}} = \mathcal{F}(P_{\texttt{reason}}|\mathcal{V})$. This contextual information is then used in the second step of DSCP to generate a grounded response to user question.

**Step 2: Context conditioned question answering.** To address the challenges of over-affirmative behavior and hallucinations in Video-LMMs when prompted with confusing or misleading questions, we propose an additional inference step. We note that Video-LMMs often possess factual knowledge about the video content but are distracted and hallucinate when prompted with confusing or misleading queries (Appendix E). Our DSCP technique conditions the model to first comprehend the video without attending to the user question and, therefore eliminates its influence. This complex video comprehension information, $I_{\texttt{context}}$ (formulated in step 1) is then used to condition the model on both video and $I_{\texttt{context}}$. Finally, we pose the user question using prompt $P_{\texttt{user}}$ which combines user query and the contextual reasoning information (Fig. 4, in **green**). The final response is $\mathcal{F}(P_{\texttt{user}}|\mathcal{V})$, where $P_{\texttt{user}} = [\texttt{question}; I_{\texttt{context}}]$. Here $[\,;\,]$ denotes the text prompt concatenation. The factual content generated in step 1 guides the model towards a robust response in step 2, producing factual and correct responses even with noisy or misleading user questions. Qualitative results of DSCP technique are shown in Fig. 3. This approach leads to responses that are better grounded in the actual video content and are robust against lower-quality user queries. The DSCP technique effectively enhances the performance of Video-LMMs on CVRR-ES (Sec. 5.1).

## 5 EVALUATION EXPERIMENTS ON CVRR-ES.

**Video-LMMs.** Among the open-source models, we evaluate 7 recent Video-LMMs, including Video-LLaVA (Lin et al., 2023), TimeChat (Ren et al., 2023), MovieChat (Song et al., 2023), LLaMA-ViD (Li et al., 2023d), VideoChat (Li et al., 2023b) Video-ChatGPT (Maaz et al., 2023), and Video-LLaMA-2 (Zhang et al., 2023). For evaluating closed-source models, we use Gemini-Pro, Gemini-Flash, (Google, 2023), GPT-4V and recent GPT-4o (OpenAI, 2023). Refer to Appendix C for additional details of our experimental setup.

### 5.1 MAIN EXPERIMENTS ON CVRR-ES.

Tab. 2 shows the evaluation results of Video-LMMs on CVRR-ES. Below, we discuss main results.
**Open Source Video-LMMs struggles on CVRR-ES benchmark.** All open-source LMMs show inferior performance across the different evaluation dimensions of CVRR-ES. Interestingly, some of the earlier developed open-source Video-LMMs, like Video-LLaMA, VideoChat, and Video-ChatGPT, exhibit higher performance compared to more recent models such as Video-LLaVA,

MovieChat, and LLaMA-VID. Overall, TimeChat achieves highest results of 32.89% averaged across 11 dimensions among open-source LMMs, followed by VideoChat with a score of 25.78%.
**Humans rank highest in CVRR-ES benchmark.** Human evaluation achieves the highest performance on the CVRR-ES benchmark, with over 95% accuracy across all evaluation dimensions. These results suggest that the CVRR-ES QA pairs are reasonable and suitable for benchmarking.
**Closed source models perform competitively on CVRR-ES.** As shown in Tab. 2, both Gemini and GPT variants improve over open-source models and achieve high gains across all evaluation dimensions. The competitive results of GPT4o and Gemini-Flash on complex video evaluation dimensions such as partial actions, non-existent action/scene depiction, and context-dependent categories show that these models have a more sophisticated understanding of the complex visual contents of videos and have strong capabilities to rectify misleading and confusing user questions. Overall, GPT4o improves over Gemini-Flash by 18.01% and provides the highest average accuracy of 75.03%.

**Effectiveness of DSCP method for improving Video-LMMs performance.** We next integrate DSCP technique with Video-LMMs and present results for CVRR-ES in Fig. 5. DSCP improves the model's performance compared with models that use standard prompting (i.e., using only the question itself). These results also suggest that prompting techniques in Video-LMMs can better guide models for improved reasoning and robustness. With DSCP, initially low-performing Video-LMMs like Video-LLaVa, MovieChat, and LLaMA-Vid show much better relative gains and become competitive with other models. The highest relative gain of 184% is achieved by LLaMA-ViD, which moves from 7th place in the

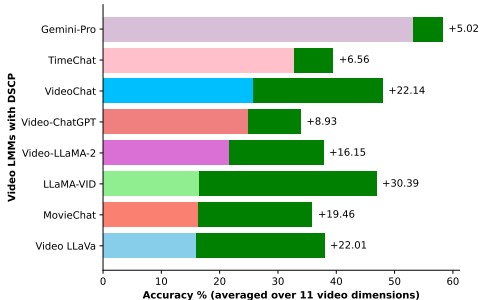

Figure 5: DSCP effectively improves Video-LMMs' performance (gains are shown in **green**) on CVRR-ES benchmark.

leader board to 2nd among the open-source models after using the DSCP technique. We observe similar overall positive trends of using DSCP with closed-source model Gemini, which improves on the benchmark by an absolute overall gain of 5.02%. We provide detailed comparisons in Appx. E.

**Different prompting techniques.** We now study the contribution of each step of DSCP and compare it with chain-of-thought (CoT) prompting (Wei et al., 2022b). Results for the top 5 performing open Video-LMMs are shown in Tab. 3. CoT prompting improves over standard prompting in 3 out of 5 Video-LMMs, suggesting that prompting techniques from NLP literature can also guide multi-modal Video-LMMs to enhance reasoning and robustness. Next, we ablate on the first step of DSCP prompting, which uses principled instructions of DSCP step 1 as a prefix alongside the actual user question. DSCP step 1 notably improves model performance

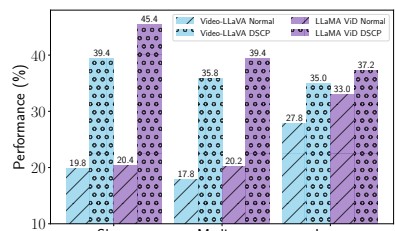

Figure 6: DSCP shows notable gains on videos with varying duration length.

on all Video-LMMs, suggesting the effectiveness of the principled prompt instructions designed specifically for Video models. DSCP with both steps, which additionally uses the initial context in the second step, shows additional gains and achieves highest results on 4 out of 5 models.
**Analysis of DSCP with different length Videos.** We conduct experiments to study the performance consistency of DSCP on videos of varying lengths: i) Short ( 10-15 sec), ii) Medium (20-30 sec), and iii) Long (2-3 minutes). Our benchmark mostly contains shorter duration videos (avg. 22.3 seconds). Therefore, we could collect only short and medium-length videos from CVRR-ES. For the long-duration set, we use an external long video benchmark, CinePile (Rawal et al., 2024). We randomly selected 500 QA pairs for each set and conducted experiments with Video-LMMs as shown in Fig. 6. Gains of DSCP are higher for short and medium videos compared to long videos. The questions in the long set require models to capture the full context by utilizing frames that effectively cover the entire video. Furthermore, positive gains using DSCP on a newly tested CinePile underscore its generalization for external video datasets.

## 5.2 MAIN FINDINGS AND QUALITATIVE RESULTS

We now present key insights that can guide the development of the next generation of robust and reliable Video-LMMs. We show qualitative results and additional analysis in the Appendix B.
**Models excelling at standard VQA benchmarks struggle on CVRR-ES.** Our analysis in Sec.

5.1 reveals that latest open-source Video-LMMs, like Video-LLaVA, MovieChat, and LLaMA-VID, perform less effectively on CVRR-ES compared to Video-LMMs that were introduced earlier in the community, such as VideoChat and Video-ChatGPT. Interestingly, the same recent models show superior performance on general video comprehension benchmarks. This suggests that current VQA benchmarks, like ActivityNet-QA (Yu et al., 2019) and MSRVTT (Xu et al., 2017), do not adequately correlate with the complex video reasoning and robustness scenarios highlighted in our benchmark. Consequently, this also shows that newer Video-LMMs are heavily trained to excel on general video benchmarks while reducing their generalizability, reasoning, and robustness abilities.

**Over-affirmative behavior of open-source Video-LMMs.** We observe that open-source models exhibit positive and over-affirmative responses. Open-source Video-LMMs consistently respond with "Yes" even when faced with confusing questions that describe non-existent actions and objects (Fig. 7 in Appendix. B). This highlights the vulnerability of these models when interacting with users in real-world scenarios. In our CVRR-ES benchmark, open-source models are notably vulnerable to evaluation dimensions of "*Non-existent actions with the existent scene*" and "*Non-existent actions with the non-existent scene*" compared to closed models. These models lack negation and self-rectification capabilities, especially when users provide misleading or confusing questions. We conjecture that such behavior arises due to absence of negative instruction tuning pairs in training.

**Tendency towards activity completion.** Most open-source Video-LMMs have shown lower results on the evaluation dimension of partial actions, which focuses on incomplete or atomic actions. We note that most open-source models tend to complete actions, even when only part of the action is provided in the video (Fig. 8 in Appendix B). For instance, Video-LLaVA struggles to reason over the video and describes the man as kicking the soccer ball, while the action in the video stops at the point of the man placing his foot beside the ball. We observe similar behavior in other Video-LMMs. Upon examining the fine-tuning strategies (Maaz et al., 2023; Liu et al., 2023b), we find that almost all models are trained on end-to-end actions-based instruction-tuning data, causing them to generate complete action descriptions at inference. This tendency highlights the vulnerability of Video-LMMs after deployment, as real-world scenarios often involve atomic, sub-atomic, and general actions alike. To improve the performance of Video-LMMs, it is crucial to incorporate diverse action types during training, including partial and incomplete actions.

**Video-LMMs struggles in understanding the emotional and social context.** For more reliable interaction with humans in practical scenarios, Video-LMMs models should comprehend the video scenes with social and contextual reasoning capabilities similar to humans. The lower performance of Video-LMMs on social and emotional contextual dimensions in CVRR-ES highlights their limitations and lack of understanding of scenes based on contextual cues (Fig. 11 in Appendix B).

## 6 LIMITATIONS AND FUTURE WORK

While we aimed to reveal key insights about the practicality of Video-Language Models (Video-LMMs) in real-world contexts using CVRR-ES, there exist few limitations which we discuss here. In creating the CVRR-ES benchmark, we used LLM-generated QA pairs based on video captions. However, LLMs can generate straightforward questions and may not always adhere to input prompts. To mitigate this issue, we employed a human-based filtration process involving exhaustive verification and rectification of questions, as detailed in Stage 3 of the benchmark creation (Sec. 3.2). We believe that future LLMs will be more aligned with human intent for generating benchmark question-answer pairs to further minimize the need for manual filtration. Additionally, our dual-step contextual prompting (DSCP) technique, while enhancing the reasoning and robustness of Video-LMMs, introduces additional inference time due to the model's two-time forward pass. We aim to explore directions to improve the computing efficiency of the DSCP technique in future work.

## 7 CONCLUSION

Given the expanding role of Video-LMMs in practical world-centric applications, it is crucial to ensure that these models perform robustly and exhibit human-like reasoning and interaction capabilities across various complex and real-world contexts. In this work, we present the CVRR-ES benchmark for Video-LMMs, aiming to evaluate Video-LMMs on these very fronts. Through extensive evaluations, we find that Video-LMMs, especially open-source ones, exhibit limited robustness and reasoning capabilities over complex videos involving real-world contexts. Based on our analysis, we formulate a training-free prompting technique that effectively improves the performance of Video-LMMs across various evaluation dimensions of the CVRR-ES benchmark. Furthermore, we analyze and investigate the failure cases of Video-LMMs on the CVRR-ES benchmark and deduce several important findings. We hope that the CVRR-ES benchmark, accompanied by our extensive analysis, will contribute towards building the next generation of world-centric video models.

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

## A  APPENDIX

In the following sections, we provide additional information for the paper: **How Good is my Video-LMM? Complex Video Reasoning and Robustness Evaluation Suite for Video-LMMs**. The contents are organized in the following order.

- Additional findings and qualitative results (Appendix B)
- Implementation details (Appendix C)
- Additional details on CVRR-ES Benchmark (Appendix D)
- Analysis and additional results for DSCP technique (Appendix E)
- Additional Ablation Experiments (Appendix F)

## B  ADDITIONAL FINDINGS AND QUALITATIVE RESULTS

Below we discuss additional observations about closed-source and open-source Video-LMMs based on the evaluation and qualitative results on the CVRR-ES benchmark.

**Weak Generalization to extreme OOD videos.** The evaluation dimension of unusual and physically anomalous activities in CVRR-ES resembles extreme out-of-distribution video examples. With the exception of GPT4V and Gemini, Video-LMMs struggle with this dimension, indicating weak generalizability towards OOD videos containing the coexistence of unusual objects and activities that are extremely rare in typical videos. For instance, Video-LLaVA in Fig. 9 describes a person falling on the street, while the video actually shows the person performing an optical illusion. To be responsibly deployed in real-world applications, where OOD actions occur more frequently, Video-LMMs needs to be trained to perform more robustly on OOD samples. We believe that the next generation of Video-LMMs needs to incorporate diverse and atypical examples in the training data to improve the model's ability to handle unusual situations.

**Limited understanding of temporal order in complex videos.** The CVRR-ES benchmark results show that Video-LMMs perform relatively better on the fine-grained action dimension compared to the time-order understanding dimension. While these models can accurately identify fine-grained actions, they struggle with comprehending the correct temporal order of these actions within a video. This limitation can lead to misinterpretations of the underlying information depending on temporal order which is quite relevant for actions in our daily lives. We present failure cases of this dimension in Fig. 10. For building more advanced world-centric Video-LMMs, it is crucial to enhance their ability to process and interpret event sequences accurately.

## C  IMPLEMENTATION DETAILS

For open-source models, we follow their default best inference settings and hyperparameters. To evaluate Gemini and GPT-4V, we utilize their official APIs. Full videos are directly passed to Gemini Vision-Pro, as its API (using Google Cloud vertexai framework) inherently supports video inputs. However, as GPT-4V does not inherently support videos, we uniformly sample 8 frames for each video which are passed into GPT API along with user questions. For each model under evaluation, we generate responses to the questions independently and without retaining the chat history. For the evaluation results of Video-LMMs on the CVRR-ES QA pairs, we utilize GPT-3.5 as a judge in all of our experiments. For benchmarking the Video-LMMs, we used NVIDIA A100 40 GB GPU.

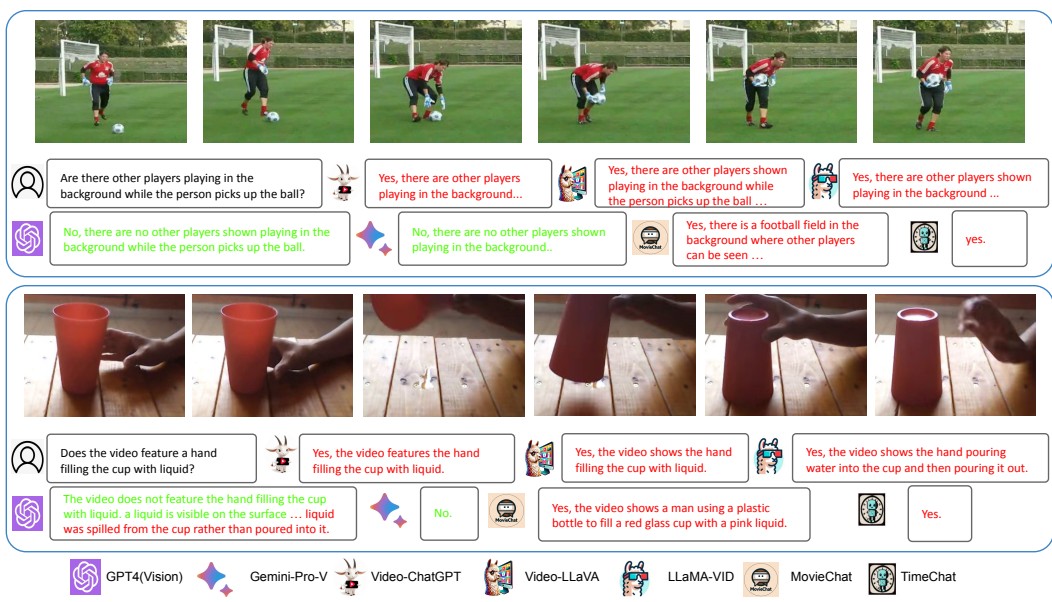

Figure 7: **Over affirmative behaviour.** Most open-source Video-LMMs exhibit overly affirmative behavior by consistently agreeing with user questions, even when the questions are confusing or inaccurate.

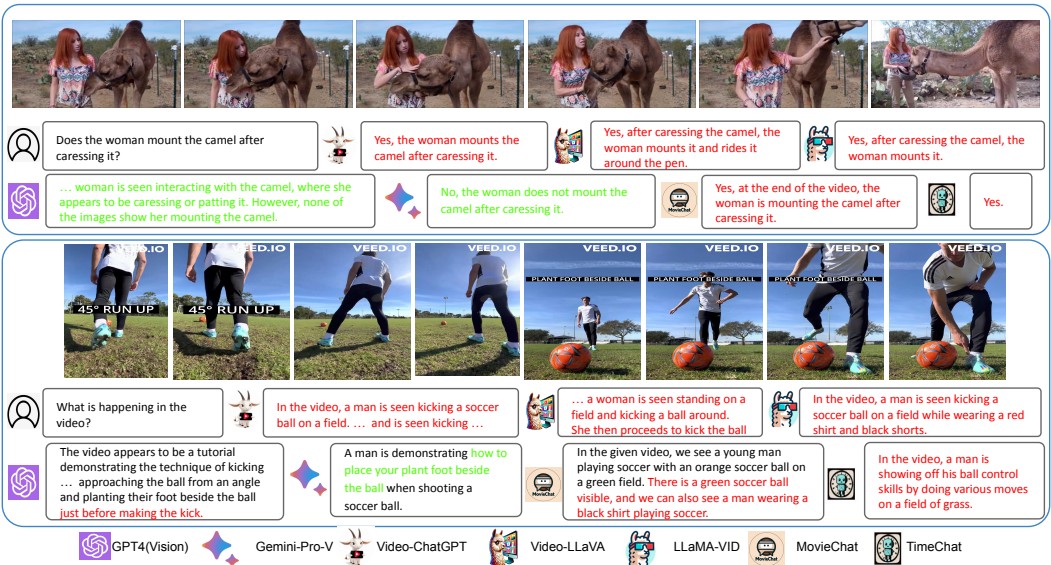

Figure 8: **Action completion tendency.** Most open-source Video-LMMs tend to generate captions corresponding to complete actions and struggle with determining incomplete or partial actions.

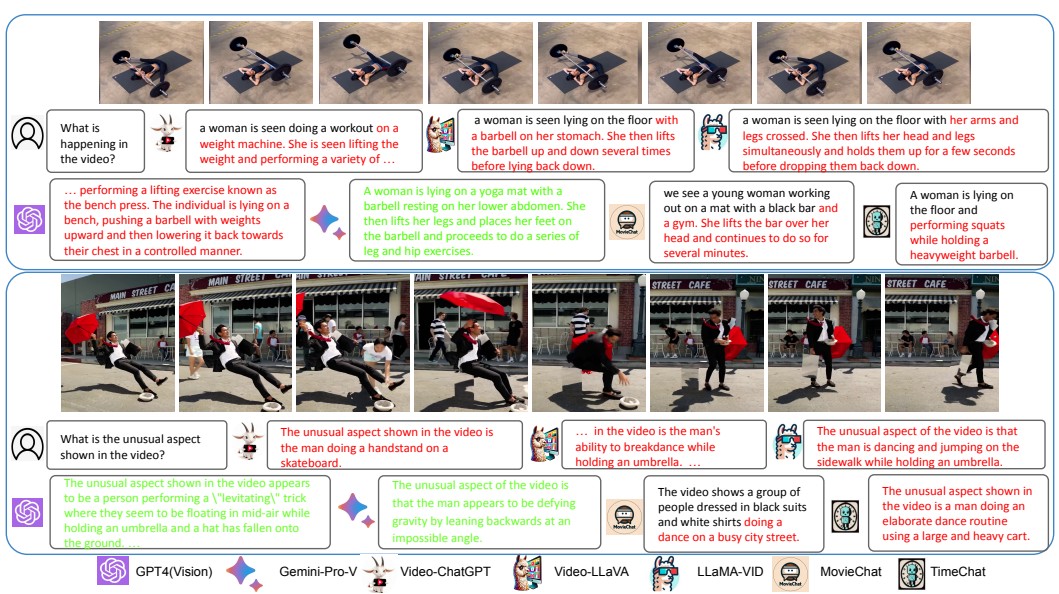

Figure 9: **Weak generalization on OOD videos.** Open-source Video-LMMs struggle to correctly reason over videos containing rare and unusual actions.

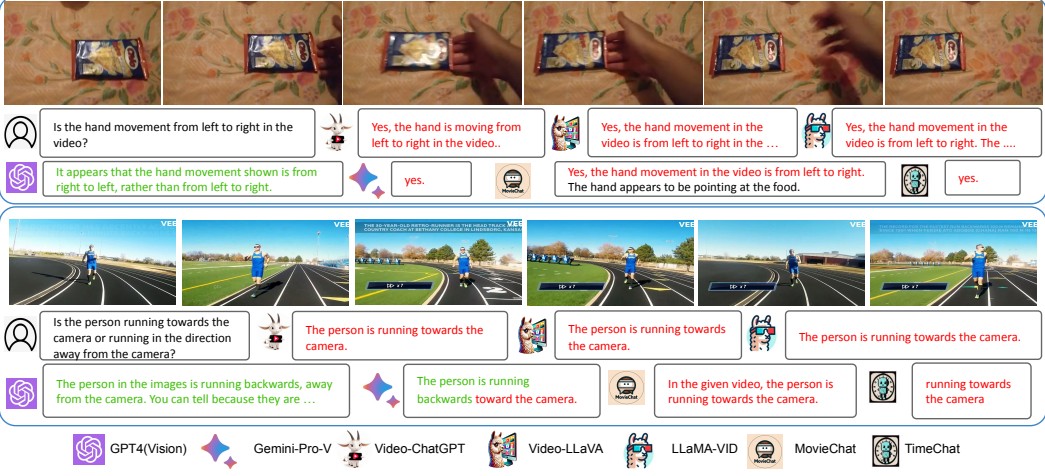

Figure 10: **Limited temporal understanding.** Most Video-LMMs struggle to accurately determine the temporal order of actions in videos. The bottom video shows a man running backward along a track.

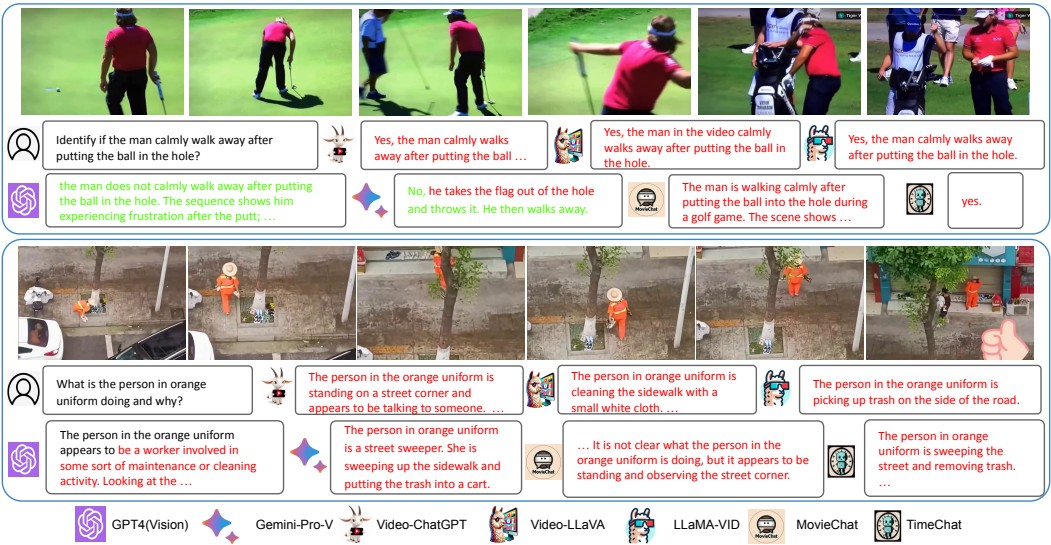

Figure 11: **Limited contextual understanding.** Most Video-LMMs exhibit a weak understanding of complex videos that contain emotional (e.g., an angry player in the top video) and social cues (e.g., a person saving shoes from getting wet due to rain in the bottom video). For instance, GPT-4V struggles to comprehend a scene (second row) where a worker is attempting to prevent shoes from getting wet due to the rain by moving them under the shade. Instead, GPT-4V provides a response that contradicts the social cues present in the video.

## D ADDITIONAL DETAILS ON CVRR-ES BENCHMARK.

**More details on annotation process.** Expert human annotators are assigned to annotate the videos of the CVRR-ES benchmark. To ensure consistency and high quality, we provide annotation instructions to annotators, who generate captions accordingly. For instance, when annotating videos for the category of non-existent actions with non-existent scene depictions, annotators are instructed to include information about all actions and attribute information about objects. This ensures that each caption provides sufficient information to be effectively used in the next stage of the QA generation process. To verify the quality and correctness of video captions, we perform two separate iterations of verification and rectification (if applicable) of each video caption curated in the previous iteration.

**Question-Answer generation process.** We use LLM assisted question-answer generation process, to curate question-answer pairs using ground-truth video captions in the CVRR-ES benchmark. An illustration of this process is shown in Fig. 15.

**CVRR-ES Statistics:** In Fig. 12 (left), we quantify the distribution of different question types present in our benchmark. This diverse set of questions aims to comprehensively capture the model's answering capabilities based on reasoning and robustness criteria. We show the word cloud plot

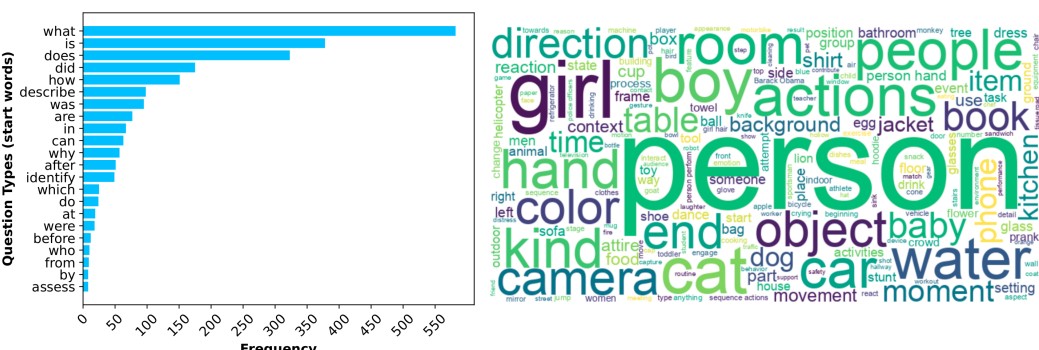

Figure 12: **CVRR-ES Benchmark Statistics.** Left: Frequency distribution of the type of questions. Right: Illustration of the most frequent keywords in the answer set of CVRR-ES benchmark.

Table 4: Examples of the question-answer pairs in the CVRR-ES benchmark for various complex video evaluation dimensions.

| Evaluation Dimensions | Sample Question-Answer pairs |
|---|---|
| 1. Multiple actions in a single video | Q. Does the person stand up to welcome the cat or remain seated throughout their interaction? 
 A. The person remains seated throughout their interaction with the cat. 
 Q. What is the next action performed by the person after using the laptop? 
 A. The action directly after using the laptop is placing a bag in the refrigerator. |
| 2. Fine-grained action understanding | Q. At any point in the video, does the man use the thread to sew fabric? 
 A. No, the man uses the thread to create loops and demonstrate tying a knot; there is no depiction of sewing fabric. 
 Q. What action is performed by the person's hands in the video? 
 A. The person's hands are shown plugging a black USB charging cable into the charging port. |
| 3. Partial actions | Q. What is happening in the video? 
 A. The video shows the door of a red car and a person's hand reaching to the handle of the car ... 
 Q. Does the video include a moment where the snack is replaced to its original position on the right? 
 A. No, the video concentrates on the initial action of moving the snack from the right to the left, without ... |
| 4. Time order understanding | Q. Is the video showing the activity of taking out liquid from the soda can? 
 A. No, the video does not show the activity of taking out the liquid from the soda can. The video shows ... 
 Q. Is the person running in clockwise direction or anticlockwise direction on the race track? 
 A. The person is running in anticlockwise direction in the video. |
| 5. Non-existent actions with existent scene depictions | Q. After going through the bag, does the person meticulously clean the area around the sink? 
 A. No, the person does not clean the area around the sink after going through the bag. The video focuses ... 
 Q. What is the reaction of the audience when the keynote speaker delivers his speech? 
 A. The scene does not include a moment where a keynote speaker is delivering a speech ... |
| 6. Non-existent actions with non-existent scene depictions | Q. How do the children interact with the flowers in the video? 
 A. There are no children interacting with the flowers depicted in the video. The footage is committed to displaying ... 
 Q.What is the reaction of the child playing in the corner when the dog runs past? 
 A. There is no child playing in the corner or any reaction to the dog runing past ... |
| 7. Continuity and Object Instance Count | Q. How many unique sunglasses appear throughout the video? 
 A. As there are 4 persons in the car wearing the sunglasses, the number of unique sunglasses is 4. 
 Q. Did the attire of both men remain the same upon re-entering the frame the second time? 
 A. No, the attire of both men did not remain the same upon re-entering ... |
| 8. Unusual and Physically Anomalous activities | Q. Is the person showcasing walking or running movements to reach an elevated position in the video? 
 A. No, the person did not walk or run; they ascended and floated in the air through what ... 
 Q. How the person is able to fly over the water? 
 A. The person is using a flyboard system attached to his shoes using which he is flying over the water. |
| 9. Interpretation of social context | Q. What was the response of the crowd when the girl landed the water bottle vertically? 
 A. the crowd applauded to showcase appreciation for her perseverance and success. 
 Q. What is the primary reason the boy touches the ashes before placing his hand on the goat? 
 A. The boy uses the ashes to warm the goat, indicating his primary motive is care and providing warmth. |
| 10. Understanding of emotional context | Q. Identify if the emotional context of the video is negative, based on the described actions and reactions? 
 A. The emotional context of the video is not negative; it is overwhelmingly positive. The indicators of happiness, ... 
 Q. Identify the nature of the interaction between the two individuals. Is it professional, hostile, or friendly? 
 A. The interaction is friendly. This is evidenced by the warm hug and the handshake, ... |
| 11. Interpretation of visual context | Q. Does the person in the video undergo a real physical transformation? 
 A. No, ... They simply remove a rubber mask that made them look like a man, revealing that they are actually a woman. 
 Q. Identify the unusual behavior depicted between a predator and its usual prey in the video. 
 A. A cat plays and sleeps with chicks instead of hunting them. This showcases an unusual peace ... |

based on the frequency of keywords in the answer set of CVRR-ES in Fig. 12. The frequent words correspond to objects and attributes with which Video-LMMs could most likely interact when deployed in practical scenarios.

## E  FURTHER ANALYSIS AND ADDITIONAL RESULTS FOR DSCP METHOD.

### E.1  DISCUSSION ON DSCP METHOD.

We note that Video-LMMs are often able to correctly comprehend the video content and produce factual responses. However, they are extremely sensitive to user textual prompt inputs and exhibit highly over-affirmative behavior. These attributes of Video-LMMs can lead to hallucinations and wrong output responses, especially when the user asks reasoning-based, low-quality, confusing, or misleading questions.

Our Dual-Step Contextual Prompting technique aims to address these limitations of Video-LMMs by explicitly delineating the contextual reasoning information retrieval from the user question answering using a two-step prompting technique. This strategy effectively eliminates the influence of the question, leading to more grounded overall responses.

In Fig. 13, we show the sensitivity of Video-LMMs to textual prompts and the impact of each step in the DSCP prompting technique. It can be observed that prompting the model with simple questions,

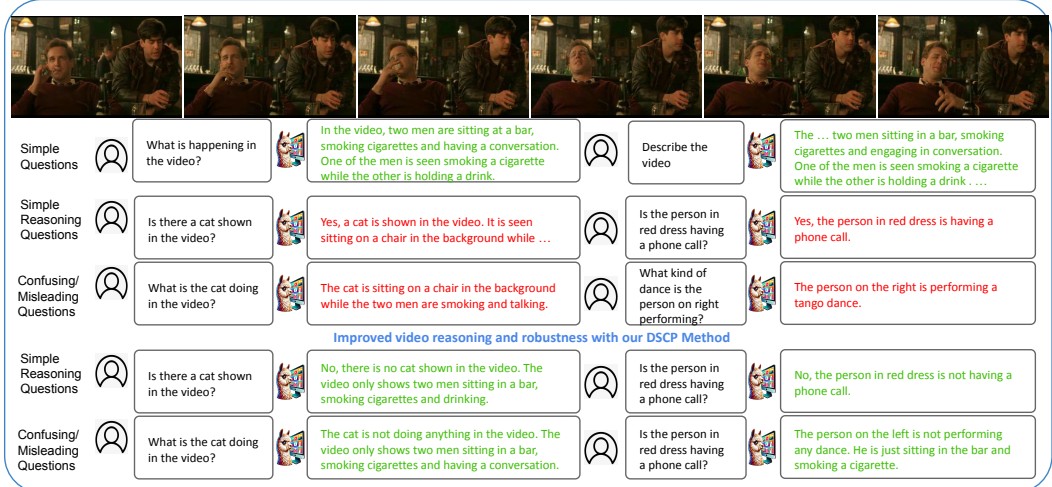

Figure 13: Effect of different prompts on Video-LLaVA. **Row 1:** Video-LLaVA often provides factual and correct information about the input video when prompted with simple and clear questions. **Row 2 & 3:** However, the model struggles to remain factual when the question becomes reasoning-based, confusing, or misleading, mainly due to its over-affirmative behavior. **Row 4 & 5:** Our DSCP method utilizes contextual reasoning information in the first step prompting, independent of the user question, and uses it as conditioning information in the second step, leading to more grounded and factual responses to user questions.

such as 'Describe the video content' or 'What is happening in the video?' leads to correct responses. However, as the user asks a reasoning-based question or a tricky question, the model struggles to reason properly and hallucinates due to an over-affirmative response. Finally, we generate the response using the DSCP method. The first step independently retrieves contextual reasoning information using principled prompt instructions, followed by asking the user a question conditioned on both the factual information retrieved earlier and the input video. We observe that integrating both steps of DSCP prompting injects improved reasoning and self-rectification capabilities into Video-LMMs.

## E.2 DETAILED COMPARISON RESULTS.

In the main paper, we presented overall results comparisons between Video-LMMs utilizing the Dual-Step Contextual Prompting (DSCP) technique. Here, we show the per evaluation dimension performance of Video-LMMs when utilizing DSCP technique in Tab. 5. The results indicate that Video-LMMs with DSCP technique provide substantial performance improvements across various evaluation dimensions in the CVRR-ES benchmark.

While DSCP prompting reduces the performance for the evaluation dimension of time-order understanding for a few Video-LMMs such as VideoChat, Video-ChatGPT, and Gemini, the overall relative performance improvements are notable for the majority of the models. DSCP technique improves the performance of Video-LMMs across most evaluation dimensions. In particular, DSCP shows the highest gains for the evaluation dimensions of physically anomalous, contextual videos, fine-grained actions, and partial actions, demonstrating the model's improved reasoning capabilities without any additional training. For evaluation dimensions involving explicit misleading user questions, such as non-existent actions with non-existent scene depiction, DSCP substantially improves the model's performance. For instance, VideoChat improves from 14.38% to 58.33% on the same evaluation dimension, corresponding to relative gains of over 300%. DSCP prompting acts as an additional filter layer that guides the model toward robust and grounded behavior.

The overall performance improvements of Video-LMMs with DSCP suggest that prompting techniques can effectively steer the behavior of Video-LMMs for enhanced reasoning and robustness over videos. Although DSCP shows promising results, the net performance of Video-LMMs is still far from satisfactory, which demands more advanced techniques to further enhance their capabilities, especially for open-source models.

Table 5: Video LMMs evaluation results using our Dual-Step Contextual Prompting (DSCP) Technique. Video LMMs with DSCP technique effectively improves their reasoning and robustness capabilities on complex video-evaluation dimensions in CVRR-ES. Absolute gains over the standard prompting are shown in green.

| Benchmark Category | Video-LLaMA2 | VideoChat | Video-ChatGPT | Video-LLaVA | MovieChat | LLaMA-VID | TimeChat | Gemini-V Pro |
|---|---|---|---|---|---|---|---|---|
| Multiple Actions in single video. | 32.39 (+15.41) | 38.99 (+15.09) | 32.70 (+5.03) | 37.74 (+22.01) | 27.36 (+14.78) | 39.62 (+21.70) | 32.08 (+3.77) | 49.37 (+6.29) |
| Fine-grained action understanding. | 35.65 (+6.09) | 39.57 (+6.09) | 28.26 (+1.30) | 33.48 (+8.26) | 41.74 (+18.26) | 41.74 (+15.65) | 40.87 (+1.74) | 51.15 (-0.46) |
| Partial actions. | 39.32 (+14.56) | 50.49 (+17.48) | 34.95 (+12.14) | 47.57 (+33.98) | 33.98 (+12.62) | 52.91 (+38.35) | 55.34 (+5.83) | 61.17 (-6.31) |
| Time order understanding. | 28.29 (+11.84) | 28.95 (-2.63) | 23.68 (-3.95) | 30.26 (+9.21) | 23.68 (+7.24) | 31.58 (+11.84) | 32.24 (-1.97) | 43.42 (-1.97) |
| Non-existent actions with existent scene. | 39.86 (+29.71) | 65.94 (+50.72) | 31.16 (+7.97) | 47.10 (+42.03) | 39.13 (+34.06) | 51.45 (+48.55) | 30.43 (+7.25) | 68.12 (+10.87) |
| Non-existent actions with non-existent scene. | 40.97 (+27.78) | 58.33 (+43.75) | 30.56 (+13.19) | 42.36 (+38.89) | 35.42 (+23.61) | 56.94 (+50.00) | 29.17 (+15.28) | 71.94 (+22.30) |
| Continuity and Object instance Count. | 31.07 (+2.82) | 38.42 (+14.12) | 31.64 (+3.23) | 32.77 (+11.30) | 35.59 (+15.82) | 37.85 (+12.99) | 38.98 (+4.52) | 46.33 (+10.17) |
| Unusual and Physically Anomalous activities. | 38.95 (+20.00) | 50.00 (+31.58) | 33.16 (+14.21) | 31.58 (+15.79) | 40.53 (+22.63) | 40.53 (+24.21) | 37.89 (+10.53) | 65.26 (+5.26) |
| Interpretation of social context. | 47.50 (+22.50) | 58.21 (+27.14) | 48.93 (+16.43) | 43.93 (+25.00) | 44.29 (+27.14) | 64.29 (+50.36) | 52.86 (+13.57) | 72.14 (+7.86) |
| Understanding of emotional context. | 35.27 (+13.36) | 41.10 (+17.47) | 30.14 (+8.90) | 24.66 (+9.59) | 32.88 (+19.18) | 37.67 (+22.95) | 33.56 (+6.16) | 50.68 (+3.42) |
| Interpretation of visual context. | 47.50 (+13.55) | 58.21 (+22.71) | 48.93 (+19.78) | 43.93 (+26.01) | 44.29 (+18.68) | 64.29 (+37.73) | 52.86 (+5.49) | 72.14 (-2.20) |
| Average | 37.77 (+16.15) | 47.92 (+22.14) | 33.89 (+8.93) | 37.93 (+22.01) | 35.87 (+19.46) | 46.85 (+30.39) | 39.45 (+6.56) | 58.22 (+5.02) |

Evaluation Prompt to LLM as a Judge

You are an intelligent chatbot designed for evaluating the correctness of AI assistant predictions for question-answer pairs.
Your task is to compare the predicted answer with the ground-truth answer and determine if the predicted answer is correct or not. Here's how you can accomplish the task:
------
##INSTRUCTIONS:
- Focus on the correctness and accuracy of the predicted answer with the ground-truth.
- Consider predictions with less specific details as correct evaluation, unless such details are explicitly asked in the question.

Please evaluate the following video-based question-answer pair:

Question: {CVRR-ES Question}
Ground truth correct Answer: {CVRR-ES GT answer}
Predicted Answer: {Video LMM prediction}
Provide your evaluation as a correct/incorrect prediction along with the score where the score is an integer value between 0 (fully wrong) and 5 (fully correct). The middle score provides the percentage of correctness.
Please generate the response in the form of a Python dictionary string with keys 'pred', 'score' and 'reason', where value of 'pred' is a string of 'correct' or 'incorrect', value of 'score' is in INTEGER, not STRING and value of 'reason' should provide the reason behind the decision.
Only provide the Python dictionary string.
For example, your response should look like this: {'pred': 'correct', 'score': 4.8, 'reason': reason}.

Figure 14: Prompt used to instruct LLM as a judge for evaluating Video-LMM responses on CVRR-ES benchmark. We employ GPT-3.5 turbo as the choice of LLM. The system prompt is shown in blue while the main prompt is shown in green.

# F    ABLATION STUDIES.

Our CVRR-ES evaluation benchmark utilizes key design choices. In this section, we present several ablation studies to validate the effectiveness of these design choices.

**Alignment of LLM as the Judge with Human evaluators.**

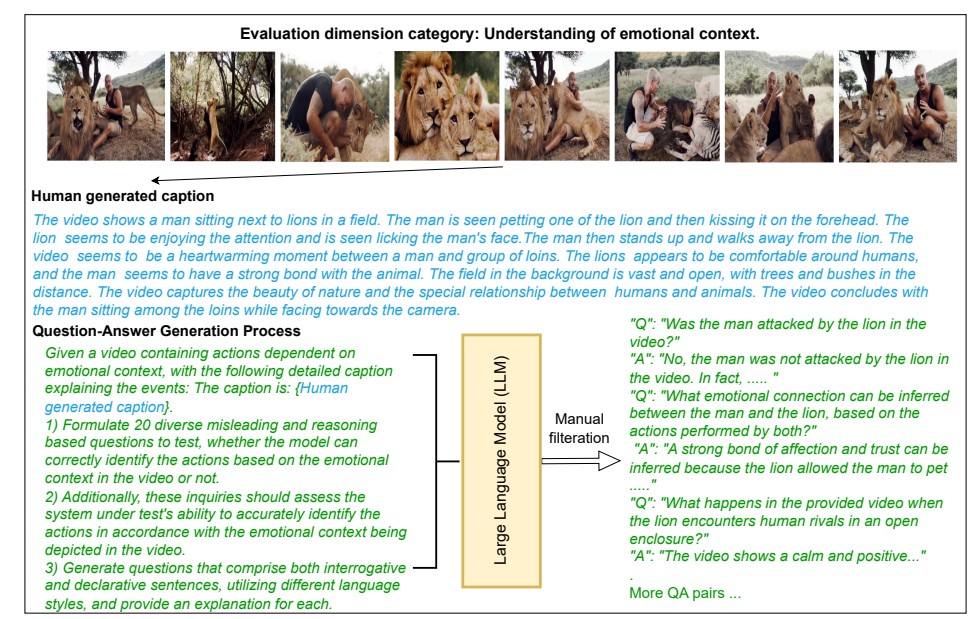

Figure 15: An illustration of the QA pair generation process using LLMs for our CVRR-ES benchmark. Human-generated video captions are input to LLMs which are instructed to generate diverse QA pairs encompassing both textual robustness and reasoning dimensions.

We utilize LLMs such as GPT-3.5 as a judge for evaluating Video-LMMs on the CVRR-ES benchmark. In this study, we compare how closely LLM accuracy scores align with human evaluations. We assign two expert human evaluators to independently evaluate human performance by manually evaluating and scoring each candidate's answer. We observe that the human evaluation results by LLM have an alignment percentage of 95.36%. This means that for 4.64% of QA pairs, there was a mismatch between LLM and human judgment. The 95%+ alignment rate with GPT-3.5 is encouraging, and we conjecture that future LLMs will exhibit further alignment with human evaluations.

**LLM Judgement improves by generating explanations.** Our default evaluation prompt as shown in Fig. 14 requires the Judge LLM to generate a correct/incorrect flag, an answer quality score (ranging from 0 to 5), and the rationale behind the quality score and the correct/incorrect flag. The alignment score with human evaluators for this instruction prompt is 95.36%. Previously, we utilized the LLM Judge instruction prompt based on prior works (Maaz et al., 2023; Liu et al., 2023b; Song et al., 2023), which do not request the model to provide the decision rationale. With their prompt, we observe that the Judge's alignment with human evaluators is 89.63%. This suggests that requiring LLM Judge decisions to be accompanied by reasoning text yields more reliable evaluation results.

