# OpenReview forum: "How Good is my Video LMM? Complex Video Reasoning and Robustness Evaluation Suite for Video-LMMs"
_ICLR.cc/2025/Conference — ICLR 2025 Conference Withdrawn Submission_

### Official Review · Reviewer_yDR5 · 2024-10-21

**Soundness:** 2
**Presentation:** 2
**Contribution:** 1
**Rating:** 3
**Confidence:** 4

**Summary:**

This paper proposes a new complex video reasoning and robustness benchmark, CVRR-ES, to assess Video-LLMs. CVRR-ES includes 2,400 high-quality open-ended question-answer pairs, spanning 214 high-quality videos, covering 11 video evaluation dimensions. This work evaluates 11 models, including closed-source and open-source Video-LLMs and establishes a human baseline. It also introduces a training-free dual-step contextual prompting method, DSCP, to enhance the performance of Video-LLMs.

**Strengths:**

- The motivation and issues discussed in the paper are valuable. By constructing the CVRR-ES to assess the reasoning ability and robustness of existing Video-LLMs on real-world videos, the benchmark is very detailed in the design of video evaluation dimensions.

- The construction of a human baseline provides a good reference for the evaluation of Video-LLMs.

**Weaknesses:**

- I express skepticism about whether the number of videos in the benchmark can achieve a robust assessment. The CVRR-ES benchmark includes only 214 videos, with the shortest video being just 2 seconds. Upon reviewing several videos from the anonymous link, I noticed a significant proportion of short videos. I question whether such short videos can adequately cover 11 categories. Moreover, current work that focuses solely on designing Video-LLMs, without specifically constructing evaluation benchmarks, provides a much larger number of assessment videos than the 214 included in CVRR-ES, for example, Tarsier [1].

- As mentioned in the previous question, the distribution of videos of different lengths within the benchmark is crucial for the assessment of reasoning ability and robustness, and the paper does not provide relevant explanations. The authors should include a table showing the distribution of video lengths across the dataset, and explain how they ensured a balanced representation of different video lengths across the 11 categories.

- In the motivation, it is mentioned that the goal is to build human-centric AI systems. Does the paper's reflection on this point merely consist of providing a human baseline? I think that offering more fine-grained visual examples would be more helpful for human-AI comparisons.

- I think that the contribution of the DSCP is somewhat overstated and lacks novelty. Such prompt engineering-based methods have already been applied in many works for data generation, model evaluation, and other stages. The introduction and ablation experiments of this technology in the paper seem redundant.

- The discussion on DSCP occupies a significant portion of the experimental analysis. I think that the current analysis provided in the paper lacks insight and does not fully reflect the value of CVRR-ES, especially in terms of human-machine comparison.

- The phrase should be "there exist a few limitations" instead of "there exist few limitations" in line 520.

- The paper does not provide prompt templates for all the closed-source and open-source Video-LLMs used, which will influence the reproducibility.

The problems discussed in this paper are valuable, but the most crucial aspects of benchmark construction and evaluation are not entirely convincing. Instead, a significant amount of space is dedicated to introducing the DSCP method. I don't think it meets the acceptance standards of ICLR yet. I will consider modifying the score based on the feedback from other reviewers and the authors' responses.

***
[1] Wang J, Yuan L, Zhang Y. Tarsier: Recipes for Training and Evaluating Large Video Description Models[J]. arXiv preprint arXiv:2407.00634, 2024.

**Questions:**

Please see the "Weaknesses".

---

### Official Review · Reviewer_SmJP · 2024-10-21

**Soundness:** 3
**Presentation:** 3
**Contribution:** 1
**Rating:** 3
**Confidence:** 5

**Summary:**

The authors introduce a new video-lmm benchmark, aiming to target specifically reasoning vs perception. They also propose a CoT prompting technique, which was shown improves performance on benchmarks.

**Strengths:**

- shows that correct prompting on video benchmarks can improve performance
- proposes a human-verified benchmark (not fully automatic pipeline), which focuses on CoT.

**Weaknesses:**

- lacks comparison to many popular benchmarks: Video-MME, PerceptionTest, MLVU, LongVideoBench, EgoSchema, TempCompass.
- note that some of these benchmarks have a higher overlap with the proposal
- Uses LLM as a judge. This makes evaluation expensive, especially at scale.

**Questions:**

- can you calculate the correlation between scores on this benchmark to existing ones (e.g., r^2) - so we can see how different the proposed benchmark is from existing ones?
- Has a multiple-choice alternative been tested, and if yes, why did you opt not to use this?
- how did you verify that this benchmark requries a 'higher' level of reasoning compared to existing benchmarks?
- does DSCP impact performance more on the proposed benchmark compared to existing benchmarks?

---

### Official Review · Reviewer_D1DJ · 2024-10-30

**Soundness:** 3
**Presentation:** 2
**Contribution:** 1
**Rating:** 3
**Confidence:** 4

**Summary:**

This paper introduces a novel benchmark suite, the Complex Video Reasoning and Robustness Evaluation Suite (CVRR-ES), designed to comprehensively assess the robustness and reasoning capabilities of Video Large Language Multimodal Models (Video-LMMs) in complex video understanding and real-world contexts. The authors also propose a training-free Dual-Step Contextual Prompting (DSCP) method, which leverages a two-stage prompting strategy to enhance model reasoning and robustness. By providing contextual cues, DSCP helps guide models toward more accurate responses and reduces misinterpretations of confusing or misleading queries. Experimental results demonstrate that current Video-LMMs often display overly affirmative responses when handling misleading or negatively framed text queries. Additionally, the experiments reveal challenges in these models' understanding of emotional and social contexts, as well as incomplete actions. The DSCP technique substantially improves model performance across these challenging dimensions.

**Strengths:**

The benchmark’s 11 evaluation dimensions are challenging and effectively assess model capabilities in complex scenarios.
The 10 principled instructions in DSCP significantly enhance the ability of Video LMMs to handle these complex tasks more effectively.

**Weaknesses:**

The contribution to the field of Video LLMs is somewhat limited. Much recent work in the community has focused on evaluating reasoning capabilities, with benchmarks like MVBench already addressing reasoning alongside a broader range of evaluation angles. Given this context, focusing primarily on complex reasoning as a contribution may have limited impact on advancing the field and may not fully meet the standards expected for ICLR.
Using GPT-3.5 as the evaluation LLM also appears somewhat outdated.
Furthermore, achieving the complex reasoning that the paper discusses requires fine-grained understanding of video content, which typically demands inputting a substantial number of video frames into the model. With a few frames, handling complex reasoning tasks becomes challenging. However, the paper does not specify the number of frames input to the open-source models, and intuitively, increasing the number of frames could enhance Video LLMs’ understanding and reasoning over video content.
Lastly, while DSCP is presented as a novel method, it is relatively basic, relying on a two-stage text prompt that appears somewhat rough and may not constitute a substantial contribution.

**Questions:**

What was the number of frames inputted for the open-source models during evaluation?
Could the authors provide a summary of the complex reasoning evaluation dimensions?  The 11 dimensions seem quite detailed; grouping some of them into broader, more general capabilities could make the benchmark easier to understand and more accessible for users, potentially aiding wider adoption.

---

### Official Review · Reviewer_sfTd · 2024-11-01

**Soundness:** 3
**Presentation:** 2
**Contribution:** 2
**Rating:** 6
**Confidence:** 4

**Summary:**

In this paper, the authors present a Complex Video Reasoning and Robustness Evaluation Suite (CVRR-ES) for assessing the performance of Video-LMMs on 11 diverse real-world video dimensions. Moreover, by evaluating 11 recent Video-LMMs, they further develop a training-free Dual-Step Contextual Prompting (DSCP) technique to boost  enhance their performance on the proposed benchmark.

**Strengths:**

1 Topic is good. Video understanding is an important problem in the multimodal research.

2 Contribution is good. Instead of investigating the exsiting comprehension abilities, the authors propose to focus on reasoning capabilities over complex videos and robustness to user prompts. Both aspects are important parts for video understanding evaluation.

**Weaknesses:**

1 The build-up of this benchmark is similar to MVBench. In Figure 1,  It is not correct that, MVbench does not contain "in the wild" and "contextual dependency".  Moreover, MVbench contains the reasoning QA. Please further make the clarification on the difference, the robustness to user prompt? More explanation should be added.

2 Dual Step Contextual Prompting is straightforward.

3 The firgures are way to small. They are not quite easy to see.

**Questions:**

Please see weakness.

---

### Note · Authors · 2024-11-20

I have read and agree with the venue's withdrawal policy on behalf of myself and my co-authors.